# Facilitating factors and barriers in help-seeking behaviour in adolescents and young adults with depressive symptoms: A qualitative study

Eline Eigenhuis[1]*, Ruth C. Waumans[1], Anna D. T. Muntingh[1], Marjan J. Westerman[2], Marlinde van Meijel[1], Neeltje M. Batelaan[1], Anton J. L. M. van Balkom[1]

1 Amsterdam UMC, Department of Psychiatry, Amsterdam Public Health Institute and GGZ inGeest Specialized Mental Health Care, Vrije Universiteit, Amsterdam, The Netherlands, 2 Department of Epidemiology & Biostatistics, Amsterdam UMC, Location VUmc and Amsterdam Public Health Institute, Amsterdam, The Netherlands

☯ These authors contributed equally to this work.
* e.eigenhuis@ggzingeest.nl

**Data Availability Statement:** Data cannot be shared publicly because according to European law (AVG) data containing potentially identifying or

## Abstract

### Objective

Despite the availability of mental health care, only a minority of depressed adolescents and young adults receive treatment. This study aimed to investigate facilitating factors and barriers in help-seeking behaviour of adolescents and young adults with depressive symptoms, using qualitative research methods.

### Methods

In-depth, semi-structured interviews with 32 participants with current or previous depressive symptoms aged 16 to 24 years using thematic content analysis.

### Findings

Our sample consisted mainly of adolescents who eventually found their way to professional help. Five main themes in help-seeking by adolescents and young adults were identified: (I) Individual functioning and well-being, (II) Health literacy, (III) Attitudinal aspects, (IV) Surroundings, and (V) Accessibility. Prompts to seek treatment were disease burden and poor academic performance. Health illiteracy negatively influenced treatment-seeking behaviour. Attitudinal aspects either hampered (shame, wanting to handle the problem oneself, negative attitudes towards treatment) or facilitated (positive attitudes towards treatment) help-seeking. Furthermore, adolescents' surroundings (school, family, and peers) appeared to play a critical role in the recognition of depressive symptoms and encouragement to seek help. Barriers regarding accessibility of mental health care were found, whereas direct and easy access to treatment greatly improved mental health care use.

sensitive patient information are restricted. The data are available from the Institutional Data Access Committee of GGZ inGeest with reference to project AFBA 14-196 (contact: datamanagement@ggzingeest.nl) for researchers who meet the criteria for access to confidential data.

**Funding:** ADT Muntingh received funding from the Netherlands Organisation for Health Research and Development (ZonMw, www.zonmw.nl) (projectnr. 430000003). The funder had no role in study design, data collection and analysis, decision to publish, or preparation of the manuscript.

**Competing interests:** The authors have declared that no competing interests exist.

## Conclusion

Facilitating factors can play a critical role in the help-seeking process of depressed adolescents and young adults, and may guide efforts to increase access to mental health care of this vulnerable age group. In particular, recognition and encouragement from school personnel and peers and easy access to care providers positively influenced help-seeking in our sample. Health illiteracy and attitudinal aspects appeared to be important barriers to seeking treatment and public/school campaigns aimed at reducing health illiteracy and stigma might be necessary to improve treatment-seeking and health care utilization in this age group.

## Introduction

Depression is a major contributor to disease burden worldwide [1, 2]. In children, depression is uncommon, with a prevalence rate below 1% [3]. However, during adolescence the prevalence increases from a lifetime prevalence of 8.4% in the age group of 13–14 years to 15.4% in the age group of 17–18 years [4]. Median 12-month prevalence estimates of depression in adolescence are similar to the rates found in adults, i.e. between 4 and 7.5% [3, 5]. Furthermore, in this age group, depression and other mental disorders are by far the most important causes of disability [6]. Besides disease burden, adolescent depression may negatively influence individual development. During adolescence, teens establish relationships, educate themselves for their working life and develop personality traits that set the tone for adulthood. Accordingly, it was found that depression in adolescents has adverse outcomes in later life, including reduced educational achievements [7–9], poor social well-being [7], higher school dropout rates [10], increased high-risk behaviours and depression-related suicide [11]. Moreover, depression in adolescence is a significant predictor for mental health problems in adulthood, including depression [12, 13], suicidal behaviour [13], anxiety disorders [14] and medical problems [15].

Effective treatment for depression is available [16], and is effective, also for adolescents specifically [3]. Nonetheless, only a minority of 15–36% of depressed adolescents receive treatment [5, 17–20], leaving at least two-thirds of adolescents untreated, resulting in a major impact on their development. Additionally, there is often a substantial delay between disease onset and initial treatment contact [21], which may prolong suffering and jeopardize a healthy development.

Increasing treatment-seeking behaviour might prevent these adverse effects. Rickwood and Thomas [22] proposed a general definition of help-seeking: "In the mental health context, help-seeking is an adaptive coping process that is the attempt to obtain external assistance to deal with a mental health concern". Previous research on help-seeking behaviour for mental health problems in adolescents has identified several barriers for treatment contact. A recent systematic review including 54 studies on barriers and facilitators in help-seeking behaviours for common mental health problems in adolescents found that the two most cited barriers were stigma and negative beliefs towards mental health services and professionals [23]. Research also shows that the desire to handle problems on one's own [24, 25], low perceived need for help [20, 25, 26], difficulty in identifying symptoms of mental illness [26, 27], perceived fear of psychotherapy, the belief that a psychotherapist would not be able to be found and financial concerns were also barriers to seeking help in adolescents [28]. While research into barriers to treatment for adolescents is readily available, few studies have addressed the role of facilitating factors. The little research that has been done shows that mental health literacy [29, 30], positive past experiences with help-seeking [31–35], social support or

encouragement from others [36, 37] and confidentiality and trust in the provider [25, 38] might facilitate help-seeking for adolescents with mental health issues. However, research on facilitating factors is scarce [23]. Previous research on both barriers and facilitators has focused mainly on adolescents with mental health issues in general. Data on facilitators for help-seeking behaviour in adolescents with depressive symptomatology is only available specifically for boys [31, 36]. The studies on facilitators that included both male and female adolescents with mental health problems used focus groups or quantitative methods to gain information on the topic [33, 38]. A downside to these research designs might be that narrative descriptions, required to be able to fully understand the specific pathways of facilitating factors in help-seeking, might be missed. Qualitative research is specifically intended to promote the growth of understanding, rather than to collect factual knowledge and causal explanations [39]. No previous research uses in-depth individual interviewing to study barriers and specifically facilitators within the help-seeking process of adolescents with depressive symptomatology.

More knowledge of facilitating factors and barriers may provide directions on how to improve access to mental health care for this vulnerable age group and can contribute to the reduction of treatment delay. In addition to adolescents we also included young adults in this study because the critical developmental stages as mentioned above are not limited to the period of adolescence but continue in young adulthood. The aim of the current qualitative study is to investigate which facilitating factors and barriers play a role in the search for professional help of depressed adolescents and young adults aged 16 to 24 years, using in-depth semi-structured individual interviews.

## Methods

### Study design

This study applied a qualitative study design by means of semi-structured interviews in order to retrieve in-depth information on possible facilitators and barriers for help-seeking. The aim was to include a heterogeneous group of male and female participants, with various levels of education, with and without treatment history, and from various ethnic backgrounds.

### Sample and recruitment

Participants were purposively sampled between March 2017 and October 2018 from educational and mental health care institutions in the Netherlands, aiming at maximum diversity. Educational institutions included secondary vocational schools and three institutions for higher education of which one university. Participating mental health care institutions were public, outpatient mental health care centres. Snowball sampling was used to include extra participants with an ethnic minority background, participants without treatment history, and males. Adolescents and young adults between 16 and 24 years of age with depressive symptoms were recruited by social workers, school counsellors and health care professionals through face-to-face invitation, after which contact details were shared with the researchers if the adolescent was considering participating. Eligible participants were contacted by telephone or email by one of the researchers to provide additional information and to invite them for the interview. All potential participants received an information letter about the study. Participants received a small reward in the form of a €10 gift voucher.

Inclusion criteria were: 1) age between 16 and 24 years; 2) depressive symptoms with a minimum score of 6 on the Dutch version of the Quick Inventory of Depressive Symptomatology (QIDS) or a history of depressive symptoms as defined by having received treatment for a depressive episode; and 3) sufficient knowledge of the Dutch or English language. The QIDS self-rated version [40] is a 16-item questionnaire measuring depressive symptoms. As we

aimed to include a diverse population in order to gain a broad view of current perceptions among adolescents, there were no exclusion criteria other than mental health problems requiring immediate treatment, including prominent suicidality, psychosis, mania or current inpatient treatment.

Of the 37 adolescents approached by the researchers who initially agreed to participate, four females eventually did not participate, due to varying reasons (stay abroad, no show, lost during interview planning, and unknown reason). One participant was excluded as she appeared to have a psychotic disorder. Data collection ended when data saturation was achieved, which was checked through three additional interviews.

## Interview procedure

Interviews were guided by a topic list, which was developed based on both literature and expert opinion. After a draft version of the topic list was created using relevant literature, it was discussed with experts from the field (including mental health care professionals and an educational counsellor, the research group and a client panel of two patients) for further fine-tuning and was adjusted accordingly. The topic list contained questions on demographic data and symptoms, open questions about reasons to seek treatment and barriers and facilitating factors in help-seeking; with subsequent items focusing on recognition or under-recognition of symptoms, wanting to deal with problems oneself, knowledge and expectation of treatment options, influence of social surroundings, stigma, practical barriers and previous experiences (e.g. '*What are facilitating factors in help-seeking*?'; *'To what extent did positive previous experiences play a role in the help-seeking process*?'). During the course of the study, the topic list was updated with new insights gathered from the data. The main adaptations concerned follow-up questions on facilitators, accessibility or practical barriers and the role of cultural influences.

The in-depth, semi-structured interviews were carried out by five female researchers (one psychologist (EE), one psychiatrist in training (RW), and three master students (MvM, AIM, MV), of whom two permanent researchers and three well-instructed interns. Frequent meetings and consultations between the interviewers guaranteed interview quality.

Interviews were conducted at the participants' location of choice, which was typically at the office of the educational or mental health care institution they attended. A non-judgemental and open interviewing style was adopted. Interviews lasted 73 minutes on average (range 38–123 minutes). The interviews were audiotaped and transcribed verbatim. All identifiable information was deleted from the transcripts, and transcripts were provided with a research number.

## Data analysis

Data collection and analysis took place in an iterative process. Data analysis was conducted in an inductive manner using thematic analysis [41], focusing on participants' perceived barriers and facilitators in treatment-seeking.

The first interviews were carefully read and then manually coded by two researchers per interview independently (EE, RW, MvM), and differences were discussed until consensus was reached. Field notes were taken during the interviews and the coding process, and incorporated in the analysing process. Subsequent interviews were coded by each interviewer, using the computer software MAXQDA 12, and coding was double-checked by a second researcher. For each interview, a summary was written, which was also checked by a second researcher.

A preliminary thematic map was developed by two interviewers (RW, MvM) based on the first independently coded interviews and discussed in a small research team (RW, MvM, AM). The thematic map was further updated and adapted after every two or three new interviews in

an iterative process by four of the interviewers (RW, EE, MvM, AIM). The main themes from the initial thematic map, differences and similarities between cases and possible explanations were then discussed amongst the coders and the research team (consisting of two psychiatrists (AvB, NB), one qualitative researcher (MW), one psychologist (AM) plus the aforementioned interviewers), and further reviewed and adjusted in subsequent meetings resulting in a final thematic map including the main themes. Data saturation was discussed and completed after 32 interviews.

## Ethical considerations

The study protocol was approved by the VU medical centre research ethics committee (reference number 2016.591). The final 32 participants provided written informed consent prior to the start of the interview.

## Findings

### Participants

Thirty-two adolescents with current ($N = 29$), or a history ($N = 3$) of depressive symptoms, were interviewed. The participants had various cultural backgrounds and educational levels. Almost all of the participants sought and eventually received professional help for their depressive symptoms. The majority of participants were currently receiving a form of treatment for their depressive symptoms. Most were referred by their general practitioner (GP) to mental health care workers at the general practitioners' office, or to more specialized treatment within a psychiatric outpatient centre. Some received treatment offered by their educational institutes, school psychologist or social worker. The data of the 32 interviews were used for our analysis. An overview of demographic variables can be found in Table 1.

### Themes

Analysis of the interviews generated five main themes with different barriers and facilitators for help-seeking. These main themes were (I) Individual functioning and well-being, (II) Health literacy, (III) Attitudinal aspects, (IV) Surroundings, and (V) Accessibility. A complete overview of the findings can be found in Table 2. The most important themes are explained below.

**1. Individual functioning and well-being.** *1.1 Academic performance*. Poor academic performance, indicated by the inability to study, bad results on tests and absence in school, were mentioned by many participants as an important prompt to seek help. As a participant stated:

'*I was at an all-time low and things weren't going well at school, and I was suffering. I wanted a solution. I had always had dreams about what I wanted to become in life and what I wanted for my future, what I wanted to reach. And I just saw that big problem, blocking the way. I just couldn't continue*' *(female, age 19)*.

On the other hand, one male participant explained (in hindsight) that he performed very well at secondary school, which blinded him and his surroundings to his depressive symptoms and caused him not seek help.

*1.2 Physical symptoms*. Different physical symptoms were mentioned by the participants as the first or most prominent symptom and a reason to seek help from the general practitioner. A 20-year-old female suffered from chest pains while in secondary school and was referred to a cardiologist, before the mental origin of the symptoms was identified. Fatigue and a

**Table 1. Characteristics of study sample (n = 32).**

| Sociodemographic Variables | |
|---|---|
| Age, mean years (SD[1]), range | 20.5 (.36), 16–23 |
| Women, N (%) | 21 (65.6) |
| **Level of Education, N (%)** | |
| Low[2] | 2 (6.25) |
| Intermediate[3] | 12 (27.50) |
| High[4] | 18 (56.25) |
| **Country of Origin Parents, N (%)** | |
| Western Europe | 20 (62.5) |
| Eastern Europe | 1 (3.1) |
| Northern Africa | 2 (6.3) |
| South America | 2 (6.3) |
| Southern Asia | 2 (6.3) |
| Western Asia | 4 (12.5) |
| Southeastern Asia | 1 (3.1) |
| **Current Severity of Depressive Symptomatology, N (%)** | |
| QIDS-SR[5] | |
| None * | 3 (9.4) |
| Mild | 0 (0.0) |
| Moderate | 16 (50.0) |
| Severe | 12 (37.5) |
| Very severe | 1 (3.1) |
| **Current Mental Health Care Use, N (%)** | |
| Social worker | 4 (12.5) |
| School psychologist | 3 (9.4) |
| Mental health care worker at GP's office | 2 (6.3) |
| Psychiatric outpatient centre | 18 (56.3) |
| Treatment finished | 2 (6.3) |
| Never had treatment | 3 (9.4) |
| **Antidepressant Medication, N (%)** | |
| Current use | 9 (28.1) |
| Previous use | 4 (12.5) |
| Never used antidepressant medication | 19 (59.4) |

[1]SD = standard deviation.

[2]Low = Primary school plus a maximum of 3 years of secondary education.

[3]Intermediate = Primary school plus 4 to 6 years of secondary education.

[4]High = Primary school plus 5 to 6 years of secondary education plus higher professional or university education.

[5]QID-SR = Quick Inventory of Depressive Symptoms-Self Report.

* = These participants were included because of their history of psychological or psychiatric treatment for depression.

suspicion of an infectious disease like mononucleosis infectiosa, and back pain were also mentioned as reasons to seek help from the general practitioner. Another secondary school student said:

> 'And then I was doing really bad. Also because I lost a lot of weight [. . .] then I stopped getting my period and then my mother thought, how can you lose so much weight? What is going on with you? Then we went to my GP. And then I told him that it's because I don't feel any hunger because I feel so depressed. Then he referred me to a psychologist' (female, age 16).

**Table 2. Main themes, facilitators (f) and barriers (b) and explanation of these factors in help-seeking for adolescents with depressive symptoms extracted from the interviews.**

| Main themes | Facilitators (f) and barriers (b) | Explanation |
|---|---|---|
| Individual functioning and well-being | Academic performance (f/b) | Poor academic performance facilitated help-seeking (f). Not noticing or not experiencing problems in academic performance hindered help-seeking (b) |
| | Physical symptoms (f) | Physical symptoms accompanying depression were often a reason to seek help (f) |
| | Mental distress (f/b) | Experiencing mental distress facilitated help-seeking (f). In others, depressive symptoms like feelings of hopelessness inhibited them from seeking help (b) |
| Health literacy | Knowledge about depression (f/b) | Knowledge (f) or limited knowledge (b) about depression |
| Attitudinal aspects | Shame (b) | Being ashamed of symptoms and dysfunctioning hindered help-seeking (b) |
| | Dealing with symptoms by yourself (b) | The idea that others are unable to help, mood is something you can only change by yourself, not wanting to become a burden to family members, and not being worthy of treatment were reasons not to seek help (b) |
| | Openness (f) | Being talkative and open (f) |
| | Attitude towards treatment (f/b) | Negative (b) or positive (f) attitude, formed by hearing negative or positive stories about mental health in the surroundings or by own previous experience |
| Surroundings | Identifying and signaling of symptoms by others (f/b) | School professionals, friends and parents noticing (f) or not noticing symptoms (f) |
| | Stigma and cultural influences (b) | Expected or perceived stigma (b) across all cultural backgrounds |
| Accessibility | Accessibility of general practitioner (b) | Embarrassment, stigma and the belief that the GP is only for physical symptoms (b) |
| | Waiting time (b) | Long waiting time (b) |
| | Effectuation of referral (f/b) | General practitioner or health care professional making sure referral is effectuated (f). Adolescent needed to effectuate referral themselves (b) |
| | Direct access to treatment (f) | Access to a school mental health worker, online applying for mental health care, having contact details of mental health care at hand(f) |
| | Reimbursement (f) | The idea that treatment will not be paid for by the insurance company after a certain age (f) |

*1.3 Mental distress.* Participants named a variety of mental symptoms as the reason to seek help. Feeling down, sleeping problems, nightmares, suicidal thoughts, auto-mutilation and anhedonia were psychiatric symptoms referred to as the reason to seek help. Some participants sought help because they felt different; as though they had lost their old selves.

Most participants mentioned that mainly severe mental symptoms, like self-mutilation or suicide planning, motivated them to seek help. One participant almost tried to commit suicide, on his way to university, and then realised something was wrong:

'*You just need to feel like shit because teenagers always feel like shit. So then I was, yes, well, yes, I was always feeling bad and stupid and bad and just.. phew. But I thought: 'That must be normal'. And then, that day [the day he almost jumped in front of a train] I was like: 'Oh, this is actually not normal' (male, age 23).*

Two participants stated that their mental symptoms kept them from seeking help. One experienced feelings of worthlessness and hopelessness, which made him question the usefulness of psychotherapeutic help. Another participant said her negative self-image made her think she was not worth seeking help.

**2. Health literacy.** *2.1 Knowledge about depression.* Virtually all participants mentioned a lack of insight in their symptoms or the severity. Many participants interpreted fatigue, brooding or feeling down as normal inconveniences that occurred during puberty, or as a part of their personality. Some noticed symptoms but compared themselves to peers with more severe symptoms. The following participant initially did not seek help because of this comparison:

'*My best friend from secondary school was also very depressed, but in such an extreme way with violent self-harm [. . .] and then I would think: 'Yes, I'm a little depressed, but look at her! I shouldn't make a big deal about my feelings*' (female, age 20).

The idea that symptoms would decrease by themselves and feelings of attitudinizing were often mentioned. Some characterized symptoms as purely physical and not mental, which caused them not to realize they needed help. In contrast, two participants stated that since they did not suffer from any physical symptoms, nothing could be wrong with them.

Multiple adolescents mentioned that they gained knowledge about depression through their education, which in turn led them to seek help. A student decided to seek help after learning about depression in a psychobiology university course. Her depressive symptoms started at age 12, but she only received help while at university:

'*And if I thought about it as in, negative thinking, or if I approached it in a more pragmatic way or something. Like yes, this is also an illness. Yes, I need to look at it as an illness and not as something I'm exaggerating or making up*' (female, age 20).

After watching a video about depression on You Tube, another participant decided to seek help. Another participant took a depression self-test online which made her decide to go to the GP. In three cases adolescents mentioned having knowledge about depression (through family members suffering from depression or education), but this knowledge did not immediately facilitate help-seeking. A university student with depressive symptoms from age 17, who sought help at age 21, explained:

'*I had my suspicions, yes, I mean, I wrote a thesis at secondary school about depression and while I was reading the symptoms of depression, I was really like: 'Oh, I'm recognizing much more of this than I would like'. [. . .] I knew what depression was but I didn't do anything with that information. [interviewer: Why not?] I didn't think that going to a psychologist or if I sought help, that that would help me, because the problems are in me and the solutions are also within me*' (female, age 21).

For most of the participants, they only became aware of suffering from depression and the need for treatment after seeking help and receiving information about depression from a mental health care professional or GP.

**3. Attitudinal Aspects.**    *3.1 Shame*. Many participants mentioned that their own negative beliefs and shame about their symptoms prevented them from seeking help. The belief that their feelings and behaviour were abnormal or signs of weakness was a frequently expressed reason for not seeking help. The following participant had experienced depressive symptoms since she was 16 years old, but only received treatment at age 22 when her academic counsellor referred her to a university psychologist. She explains:

'*Well. . .since I was 16 my GP referred me to a psychologist, but at that age I thought: 'Who goes to a psychologist? I'm not crazy*' (female, age 22).

Shame concerning academic malfunctioning and the belief that they were failing in life were also reasons mentioned for not seeking help. Two participants said that weakness makes you an unattractive person and this belief was the reason not to seek help.

*3.2 Attitude towards treatment*. Two participants explained that their negative attitude towards treatment kept them from seeking help. In both cases, this negative attitude was

formed by hearing negative experiences about mental health treatment from friends. One of these participants explained:

'*I just heard many negative experiences from people around me that had been treated in mental health care institutions*: '*Yes the help is bad*' *and* '*It's not helping me at all*' *and I never heard anything positive about mental health care, so as far as mental health care was concerned, I just thought it was best for me to stay far away from it*' (female, age 20).

In contrast, many participants mentioned positive attitudes towards treatment as a facilitator for seeking help. Multiple participants mentioned that having previous positive experiences with mental health care for other mental symptoms made it easier to seek help. One participant had positive expectations about mental health care because her father was a psychologist, she received help from mental health care shortly after the onset of symptoms:

'*Yes, it's like seeing psychological help as something obvious. Just like a doctor is able to help you, a psychologist might also be able to help*' (female, age 18).

Some participants noticed positive effects of mental health care in significant others:

'*My girlfriend, she also went through some things, so she also had issues. And I notice with her that when she speaks to her psychologist, it relieves her, I think. I guess it's helping her. Even though she doesn't talk much about it to me. [. . .] For me, that was one of the reasons why I thought a psychologist might also be helpful for me*' (male, age 21).

**4. Surroundings.**  *4.1 Identification of symptoms by others*. Participants mentioned that teachers and mentors frequently noticed changes in behaviour like being quiet or unhappy in class, absence or bad results at school, which often led to a referral to a GP or mental health professional. One student said:

'*My absence at school, they asked me why that was and then I explained that I had some problems, that I was not always feeling well and fit enough to go to school. [. . .] and then they said that.. they asked if I would like to speak to someone about my problems, then I said*: '*Yes, I would like to try that*" (male, age 21).

Another participant stated that one of her secondary school teachers noticed that the contents of her poems were dark and gloomy. The teacher contacted the mentor and parents of the participant, after which the parents made an appointment with the GP. The participant was annoyed at first, but thankful for afterwards for signalling her problems. Another participant stated that a secondary school teacher noticed cuts on his arm, contacted the mother of the student, after which an appointment at the GP was made.

Quite a few participants mentioned that friends and parents also recognised depressive symptoms and advised or motivated participants to seek help from a GP or mental health professional. In a few cases, friends of the participants played a mediating role in recognising mental problems and informing teachers and mentors at school.

*4.2 Inadequate identification and signaling of symptoms by others*. In some cases participants mentioned that teachers and mentors did not notice any signs of mental suffering; especially when academic results were good. One participant stated that while he was absent, the university did not signal any problems or take action:

'*Quite often I didn't show up at my study groups, missing deadlines, those kinds of things. [. . .] University doesn't even care how you are doing on a personal level, as long as you get enough points [. . .] otherwise you're thrown out. That's the only thing that matters to them. It's not a humane environment anymore [. . .] the human support of students is failing dramatically. It is really, really, really a dire situation*' (male, age 23).

A 19-year-old female, in the first year of higher professional education stated that the educational institution noticed problems and she was repeatedly sent to talk to the dean about her academic results. She would have preferred teachers to ask questions about her mental wellbeing, and she would have liked to be referred to the school psychologist instead of the dean. Similarly, another female participant spoke to teachers in secondary school about her problems and low mood but they did not support her in seeking help.

In identifying depressive symptoms, compared to teachers and mentors, parents were mentioned by the participants much less often. Some participants mentioned that their parents did not seem to be aware of any mental problems, mental or otherwise. Some parents recognized symptoms of mental illness and encouraged help-seeking, others downplayed the symptoms or did not motivate their children to seek help. Some participants with a non-western background explained that limited knowledge about mental health within their families made help-seeking more challenging. The following quote comes from a girl that eventually sought help by herself, but was not facilitated by her family members:

'*I come, you know, from another cultural environment, so that is also a factor. [Interviewer: What's your cultural background?] Arabic culture, so.. Iraq. I was born in Iraq, and there.. people didn't grow up with psychological health care, so they will not say: 'Oh, if you have psychological trouble, you need to do this. . .'. So it needed to come from within myself. [. . .] I don't want to say that psychological issues are not accepted, because it's definitely accepted, but maybe they don't take it super seriously. I think*' (female, age 22).

In some cases, participants stated that their parents' own experience with mental health issues helped them to recognise depressive symptoms in their children.

*4.3 Stigma and cultural influences.* Many participants, from all cultural backgrounds, mentioned hiding their symptoms or not talking about them because of expected or perceived stigma. The fear of others having pity on them or being bullied was mentioned by multiple participants. Many also thought that if they had talked about their symptoms they would not have been understood by others or others would have thought they had failed in life. A participant with Dutch parents explained how she perceived stigma within her surroundings:

'*I felt ashamed about my problems and about visiting my school counsellor, especially towards my parents, but I also didn't dare to speak about it with my peers. I think I was afraid of their judgement: 'She can't do it on her own, she needs help, she is weak, she is such a bad person', you know?*' (female, age 20).

While some participants with Dutch parents mentioned stigma in their families, all participants of non-western origin spoke about stigma in relation to their cultural background. Three participants with parents from Western Asia emphasised the family belief that problems should be kept within the family. Two participants originating from Southern Asia and one from Western Asia explained that depression is not a term used within their communities and their parents were not open-minded about the subject. The following participant (born in the Netherlands, with parents originally from Southern Asia) explained his father's view of psychological help:

*'Yes, it's a real pity that everything is so concealed, that people don't talk about it. There is definitely some kind of taboo where I come from, for example on seeking help from a psychologist. [Interviewer: did this influence you in any way?]. Yes definitely. My father was also like: 'You will not find a job' and stuff. He meant in medical school. He said: 'Then they will ask for your data or something, and then you will never be able to find a job'. [. . .] They have a very black and white view on psychologists. . . and on psychological treatment. Well, they think you will be referred to a psychiatric hospital right away. Yes it really works like that over there. [. . .] For a long time I did not go to a psychologist because I also thought that I needed to deal with things by myself. And seeking help. . . I also didn't understand how that would possibly help me' (male, age 21).*

A participant originating from Northern Africa said mental health stigma plays a big part in North African society, but found good support within her family. Another participant, born in the Netherlands and with parents originating from Northern Africa, did not feel understood by her parents but stated that stigma is not something specific to their culture:

*'It's not the case with everybody. There are also enough North African women that do understand psychological issues, and will help you. Others don't, do you understand? I think it has to do with the individual. [..] There are people out there that do understand like: 'Hey, my daughter needs help, I'm going to find help for my daughter, I'm going to do it, I'm going to support her, so that she will get better' (female, age 23).*

**5. Accessibility.** *5.1 Perceived accessibility of the general practitioner.* Many participants experienced a barrier in approaching their GP. Embarrassment, stigma and believing the GP only treats physical symptoms were reasons mentioned for not going to the GP with depressive symptoms.

*'I thought it was such a stupid idea to go to my GP and say: 'So. . . I want to die.' Yes that seems super weird to say [. . .] yes, the GP is there for normal symptoms, the ones you can see, if you know what I mean. So when you have a painful hand, or something. On a physical level' (male, age 18).*

*5.2 Effectuation of referral.* Many participants in our sample did not in the end receive treatment because they had to contact the psychologist or mental health care institution themselves (often by telephone), which was a barrier to them. Some participants commented that they had needed someone to push them to effectuate the referral. As a consequence, one participant appreciated his involved GP, who monitored the trajectory:

*'My GP was immediately like 'OK, I will send you to my mental health care nurse and then I will see you in two months, to discuss how you are doing'. So he kept his communication lines short [. . .] So I appreciated that. Because I do have the idea. . . I'm a person that can delay things quite often. So I like to have someone who pushes me a bit' (male, age 23).*

*5.3 Direct access to treatment.* Many participants mentioned that they found some form of help because their school offered assistance or treatment. Reported facilitators were: the possibility to receive support from a school social worker or school psychologist; a routine appointment with the school counsellor; or an active inquiry about well-being by a teacher or mentor when grades were deteriorating. In some cases, the school referred to a psychologist or strongly

recommended the participant to seek help. In one case, the educational counsellor was also a psychologist and treated the student herself:

'*My academic counsellor in my political sciences course, was both an academic counsellor and a psychologist. So I saw her for two years*' (female, age 22).

One participant easily contacted a health professional because signing up was simplified by an online form and a telephone call from the institution the next day. One participant mentioned easy accessibility since she had direct contact details available from previous mental health care appointments. In addition, one of the participants had direct access to care as he was treated by his father. Another female was referred to a psychiatrist via her father.

Table 3 contains summaries of two interviews, demonstrating the complex process of barriers and facilitators in help-seeking.

## Discussion

This qualitative study aimed to investigate facilitating factors and barriers for formal treatment-seeking in adolescents and young adults with depressive symptoms. This study adds to the literature because contrary to previous studies focusing primarily on barriers [24–26, 42, 43] there was specific attention for facilitating factors, yielding valuable additional information on adolescents' and young adults' pathways to professional mental health care.

**Table 3. Summary of two interviews showing the complex process of barriers and facilitators in help-seeking.**

| Brian*, male, age 23, Dutch parents |
| --- |
| In the case of Brian it took many years to receive help after the onset of symptoms. |
| In high school, as a result of stress and bad grades, he began feeling down. He thought what he was feeling was just part of being a teenager. He says, now, that he didn't recognise what he felt as depressive symptoms but also that he didn't want to acknowledge those symptoms because he thought positive thinking would make the symptoms go away. When he had his first suicidal thoughts, he spoke to his family about his low mood. His mother was worried and got stressed, but did not encourage him to seek help. After a while he took the initiative to see his GP about his feelings and the GP gave him a psychologist's telephone number. He didn't call because he was temporarily feeling better. His parents weren't actively involved in his help-seeking process. After this, he stopped mentioning the issue and his family and friends didn't ask either. He never discussed it with anyone, finding the subject too intense and thus to be avoided. A year later he was still underperforming at school and nothing had really changed. He began reflecting more about his situation. Talking to his sister also contributed to him realizing that he wasn't feeling any better than a year before. It had been a very stressful year, including problems in his relationship. Altogether, this made him seek help on his own initiative. His parents were supportive. He called the psychologist's number that the GP had given him the previous year. With hindsight, he thinks the GP should have done more and blames him for the delay in seeking psychological help. |
| Imane*, female, age 23, parents originating from Northern Africa |
| In the case of Imane, her family and school played a crucial role in getting her to receive help for depression. |
| She started feeling down and not doing well at school at the age of 14. She began eating more when she was feeling down and gained weight as a result. She started skipping school and teachers began asking if she might be depressed, but she didn't address the problem. After flunking a class, she stopped school and started working. With hindsight, she says this was because she didn't know how to seek help and this was her attempt at decreasing the symptoms. She says that she might not have wanted to see her own depression because she saw depression as something very severe and not treatable. She says that within her culture [Islamic culture] people don't talk about depression, but in her home they do. Her mother (also depressed in the past) was very supportive. She spoke to her boyfriend and mother about how she was feeling. They told her to go to the GP. Her GP only focused on weight gain and gave her a flyer about physiotherapy. She didn't complain because she thought that her depressed feelings were part of her personality and she had to live with them. After speaking to another friend, she decided to take care of herself and start school again. She discussed her symptoms with her school counsellor and got a referral to a psychologist at the age of 17. |

*Names of participants are fictitious.

Our study shows that help-seeking in adolescents with depressive symptoms is often a complex process with multiple interacting factors. We identified five main themes, which could either impede or facilitate help-seeking. Impairment or deterioration in individual functioning and well-being–such as poor academic performance, physical symptoms, and mental distress–was often a prompt for help-seeking, whereas good academic performance despite depressive symptoms could be a hindrance. Health literacy was one of the pivotal aspects in help-seeking; many adolescents reported a lack of knowledge about depression and treatment possibilities, and it was only after they had gained knowledge about depression that they came to understand their symptoms, which was an important factor in acknowledging a need for help. Attitudinal aspects (such as shame, wanting to deal with symptoms by oneself, and negative perception of treatment) hampered the treatment-seeking process; however, openness and a positive attitude towards professional care were identified as facilitators. Furthermore, the surroundings; parents, peers, and school personnel in particular played an important role in help-seeking. While recognition and motivation by important others facilitated help-seeking, expected or perceived stigma impeded the willingness of young men and women to seek treatment. Lastly, barriers regarding accessibility of mental health care were mentioned, while acknowledging that direct and easy access to treatment significantly improved mental health care use.

Our finding that impairment or deterioration in individual functioning and well-being prompt treatment-seeking is in accordance with previous research in adolescents [24, 32, 44] and adults [45, 46], although there are studies in adults that indicate otherwise [47, 48]. Furthermore, previous research found that adolescents and young adults with increased suicidal ideation [38] or higher levels of depression [49] communicate a lower tendency to seek treatment. In our sample, two participants reported that the nature of their depressive symptoms (feelings of hopelessness and worthlessness) deterred them from seeking help. Thus, it is dependent on the severity and type of depressive symptoms, and the interplay with other factors, whether this is a facilitating factor or a barrier to help-seeking.

Health illiteracy and problems with symptom recognition–a major barrier in help seeking in our sample–was also found in other studies in children, adolescents, and young adults [23, 25–27, 50, 51], as well as in the adult population [45, 46]. Our study showed that for certain participants, increased knowledge about depression led to help-seeking. Correspondingly, research by Wright and colleagues [52] showed positive effects of a mental health awareness campaign for young people, leading to increased mental health literacy, a small increase in help-seeking behaviour, and a reduction of barriers to treatment.

Our finding that attitudinal aspects, including shame, positive or negative attitudes towards treatment, and wanting to deal with problems oneself influence treatment-seeking, has been found previously both in adults and youth [18, 23–27, 31, 32, 38, 45, 47, 49–51, 53, 54].

Furthermore, our findings show the importance of adolescents' surroundings–family, peers, and school personnel in particular–who often played a crucial role both in the recognition of depressive symptoms and in active encouragement to seek treatment. In fact, for many participants in our sample, encouragement by school, parents, and peers was the prompt to search for help. These findings are consistent with previous studies [19, 23, 24, 31–33, 38, 49, 53, 55] that also stress the role of parents, teachers, peers, and social surroundings in identifying symptoms and referral, both in adolescents and adults. Our study, however, extends the existing literature in several ways. While many studies endorse the important role of surroundings, this study describes the specific influence of family and school in more detail. For instance, our study found that in some cases parents had a negative influence on problem identification and treatment-seeking behaviour, since some participants mentioned that parents were unaware of their mental symptoms, had limited knowledge about mental health, or even

downplayed the problem. Furthermore, this study gives more detailed insight into the important role of the school. Although a few participants reported under-recognition of symptoms by their school, many of our participants stated that school personnel made them aware of their symptoms and the severity of their situation. In addition, schools' involvement comprised three other major aspects: (i) increased health literacy from information on depression obtained at school or university; (ii) assistance in finding professional treatment; and (iii) readily accessible basic support or treatment from school psychologists and social workers.

In addition to the above mentioned influences of social surroundings, expected or perceived stigma from others had a potential negative effect on help-seeking in our sample. Adolescents seem to be particularly vulnerable for perceived stigma, compared to adults [56]. Another interesting finding is that all adolescents in our sample with a family history of migration named the role of stigma in relation to their cultural background, compared to only a subgroup of Dutch participants. This seems in accordance with previous research, that found relatively higher vulnerability to stigma in (certain) adolescent and adult ethnic minority groups [20, 56, 57]. Thus, stigma seems to be a universal hindrance in treatment-seeking, but might be even more so in ethnic minorities. Although qualitative research is not the appropriate method to draw firm conclusions from observed frequencies, this is still an interesting finding that should be further investigated in the future.

Lastly, an important finding was the variance in accessibility of mental health care experienced in our sample. Although mental health care in the Netherlands is completely covered by an obligatory health insurance (after referral by a GP), problems concerning access to treatment were frequently mentioned and the pathway to mental health care appeared to be problematic and demanding in some cases. One main difficulty was the proactive role expected from the adolescent in arranging an appointment with a therapist after referral by a GP. This was potentially a barrier and some adolescents never succeeded in getting mental health care. Although accessibility problems were previously reported, most studies provided only limited information on this subject, mainly due to a quantitative study design [18, 24, 26, 43, 49] or a small sample in a rural area [42]. Due to the qualitative study design, our study contributes to existing literature by giving more detailed insight into the reasons and precise mechanisms of these access barriers. For instance, in our sample, all actions from the adolescent's social surroundings that enabled direct access to treatment or shortcuts to care were facilitating. This included monitoring of the referral process by the GP or a family member, direct access to school personnel offering basic support or treatment, and facilitated access to health care professionals because of previous treatment or family connections with a care provider. Accordingly, research showed that access to mental health support at school indeed increased the likelihood of receiving basic mental health care, without influencing the amount of care received outside of school [58]. Apparently, getting mental health care can be a tough process for depressed adolescents and young adults, and any simplification of the process may have a positive effect.

## Strengths and limitations

An important strength of our study is the use of in-depth interviews. Unlike quantitative data, this study design gives the opportunity to thoroughly investigate individual processes of help-seeking behaviour and interactions between the different impeding and facilitating factors. Furthermore, in addition to previous qualitative studies focussing on barriers [24–26, 42, 43], our study paid specific attention to facilitators of the help-seeking process as well. Another strength of the study is the, for qualitative research standards, sizeable sample consisting of 32 participants who were recruited in various contexts (mental health care, general population, and schools), resulting in a sample with participants from various ethnic backgrounds, of

different ages and levels of education. Furthermore, we included adolescents both with and without a treatment history.

Previous research has shown that the literature on help-seeking behaviour comprises a variety of studies with different methodology, focus and outcomes, hampering direct comparison [22]. In order to improve conceptual consistency, Rickwood and Thomas [22] presented a framework consisting of five aspects of help-seeking behaviour: Process (comprising the aspects of Orientation, Intention, Behaviour), Timeframe (i.e. past/next 4 weeks, past/next 12 months, Ever), Source (Formal help, Semi-formal help, Informal help, Self-Help), Type of help (Instrumental, Information, Affiliative, Emotional, Treatment) and Concern (General distress, Specific symptom types). Although we did not include this framework in the design of our study, our study can be categorised as focusing on all three Process aspects (Orientation, Intention and Behaviour) of help-seeking, in a broad time frame (Ever), from Formal help sources (Source), in participants with a specific syndrome type, i.e. depressive symptoms (Concern). Information on all types of help (Type) was gathered, with a specific focus on information and treatment. Future studies on help-seeking may use a conceptual framework such as described by Rickwood and Thomas [22] to increase homogeneity in study designs and comparability of the results.

A limitation of this study is the relatively small number of low-educated adolescents compared to high-educated participants. In addition, the number of untreated participants was relatively small, despite efforts to include more participants without a treatment history in the study. This could have resulted in an underrepresentation of barriers and facilitators that are specific to these subgroups. The findings of this study are most representative for females, adolescents from Dutch descent, drug-naïve adolescents and adolescents (eventually) seeking help from psychiatric outpatient centres. Future research could specifically focus on adolescents that did not receive professional help and low-educated adolescents to explore whether other processes play a role within the help-seeking process of these specific subgroups. Moreover, the interviews were conducted by five different researchers, risking a lack of continuity in the interviews. However, frequent discussions and meetings between the interviewers were held to guarantee interview quality, and having different views on the interviews and data may also have enriched data collection and analysis. Another limitation is the limited information that was gained on participants' backgrounds, depression and anxiety scores at time of illness, or demographic variables, impeding thorough analysis of the impact of these factors on perceived barriers and facilitators.

## Practical implications

Our study has several practical implications for facilitating treatment-seeking in depressed adolescents and young adults and gives indications for what could be improved to optimise this process.

To reduce barriers related to health illiteracy and stigma, school or public campaigns could be organised to improve health literacy and diminish the stigma of depression [52]. In addition, these campaigns could encourage young men and women to easily contact possible gatekeepers (e.g. the GP or school contact person) to help assess their symptoms and the need for treatment. Previous research has found promising effects from education and awareness programmes on mental health literacy, help-seeking behaviour or intention to seek help and stigma reduction [59–63], using different methods including psycho-education and (video) presentations with former patients.

With regard to problem recognition and encouragement to seek treatment, parents, peers, and particularly school personnel can be made aware of their crucial role in identifying

depressive symptoms in adolescents and young adults and consider discussing their concerns with them. Subsequently, they could actively encourage the adolescent to seek treatment if deemed necessary. Peer leading interventions have been developed previously and shown a positive effect [61] on referring suicidal peers for adult support. School personnel should be aware of signs of academic malfunctioning and be encouraged to discuss the student's wellbeing when school results are worsening.

Concerning accessibility to treatment, general practitioners could monitor the referral of adolescent and young adult patients and check for follow-up, as this seems facilitating in this population. Mental health care institutions and therapists are encouraged to improve accessibility of care, e.g. by offering possibilities for online registration or easy telephone or chat contact, or actively reaching out to referred adolescent patients. Furthermore, in our sample waiting time appeared to be a problem. This is in accordance with previous research [62, 63] and indicates that the number of young adults in need of treatment exceeds the available treatment resources. This problem needs particular attention from policy makers and governments, and suggests that rearrangement of treatment resources for youngsters might be necessary. One of the most promising approaches to facilitate access to care seems to be the provision of basic mental health care in schools [58, 64]. This could be considered a form of stepped care, where school counselors and psychologists function as primary support for distressed adolescents.

## Conclusion

This study found five main themes that play a role in adolescents' and young adults' help-seeking behaviour: individual functioning and well-being, health literacy, attitudinal aspects (including shame, wanting to deal with the problem oneself, and attitudes towards treatment), influences from social surroundings, and accessibility of care. Interventions aimed at improving health literacy and decreasing stigma, stimulating the positive effect of social surroundings on symptom recognition and encouragement to seek help, and facilitating accessibility to mental health care may further enhance help-seeking behaviour and reduce barriers to treatment for adolescents and young adults.

## Acknowledgments

We would like to thank Annemiek Marschalk and Milly Vriens for their contribution to this research. We thank all the adolescents and young adults for their willingness to participate in this study and share their personal stories.

## Author Contributions

**Conceptualization:** Eline Eigenhuis, Ruth C. Waumans, Anna D. T. Muntingh, Marlinde van Meijel, Neeltje M. Batelaan, Anton J. L. M. van Balkom.

**Data curation:** Eline Eigenhuis, Ruth C. Waumans.

**Formal analysis:** Eline Eigenhuis, Ruth C. Waumans, Anna D. T. Muntingh, Marjan J. Westerman, Neeltje M. Batelaan.

**Funding acquisition:** Ruth C. Waumans, Anna D. T. Muntingh, Neeltje M. Batelaan.

**Investigation:** Eline Eigenhuis, Ruth C. Waumans, Marlinde van Meijel.

**Methodology:** Eline Eigenhuis, Ruth C. Waumans, Anna D. T. Muntingh, Marjan J. Westerman, Neeltje M. Batelaan.

**Project administration:** Eline Eigenhuis, Ruth C. Waumans, Marlinde van Meijel.

**Software:** Eline Eigenhuis, Ruth C. Waumans, Anna D. T. Muntingh.

**Supervision:** Anna D. T. Muntingh, Marjan J. Westerman, Neeltje M. Batelaan, Anton J. L. M. van Balkom.

**Validation:** Ruth C. Waumans, Anna D. T. Muntingh, Marjan J. Westerman, Neeltje M. Batelaan, Anton J. L. M. van Balkom.

**Writing – original draft:** Eline Eigenhuis, Ruth C. Waumans.

**Writing – review & editing:** Eline Eigenhuis, Ruth C. Waumans, Anna D. T. Muntingh, Marjan J. Westerman, Marlinde van Meijel, Neeltje M. Batelaan, Anton J. L. M. van Balkom.

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
