## [Decision Letter · Decision Letter 0]

15 Oct 2020

PONE-D-20-29462

Facilitating factors and barriers in help-seeking behaviour in adolescents and young adults with depressive symptoms: a qualitative study

PLOS ONE

Dear Dr. Eigenhuis,

Thank you for submitting your manuscript to PLOS ONE. After careful consideration, we feel that it has merit but does not fully meet PLOS ONE’s publication criteria as it currently stands. Therefore, we invite you to submit a revised version of the manuscript that addresses the points raised during the review process. Both reviewers see the importance of your study, however they several made suggestions for improvements.

We look forward to receiving your revised manuscript.

Kind regards,

Therese van Amelsvoort

Academic Editor

PLOS ONE

Journal Requirements:

Reviewers' comments:

Reviewer's Responses to Questions

**Comments to the Author**

1. Is the manuscript technically sound, and do the data support the conclusions?

Reviewer #1: Yes

Reviewer #2: Partly

2. Has the statistical analysis been performed appropriately and rigorously? 

Reviewer #1: N/A

Reviewer #2: N/A

3. Have the authors made all data underlying the findings in their manuscript fully available?

Reviewer #1: Yes

Reviewer #2: Yes

4. Is the manuscript presented in an intelligible fashion and written in standard English?

Reviewer #1: Yes

Reviewer #2: Yes

5. Review Comments to the Author

Reviewer #1: Thank you for the opportunity to review your paper. This paper is qualitative study using interviews to study the help-seeking behaviours of young people with depression. The paper is well written, the findings are timely and relevant, and it contributes well to the literature.

Interview procedure

1. were any honoraria or remuneration provided to participants?

Data analysis

2. How have the quotes been selected? Have quotes been select to represent a wide range of participants? The reason I ask is because the demographics table indicates a large number of participants whose parents are from the Netherlands, but many of the quotes indicate otherwise, especially in the context of the cultural influences theme. Perhaps another quote could be added in this section from someone representing the Netherlands, given the large proportion.

Results

3. Qualitative is typically referred to as "findings" rather than "results"

4. Themes section: "personal themes" .. could you clarify what is meant by this? Perhaps this could also be clarified by the following table (2).

5. Section on physical symptoms: "a 20-year old female secondary school.." I'm wondering if this is correct given the age?

6. The themes could be strengthened by using (and adding to) a conceptual help-seeking framework (eg, Rickwood et al 2012) so that the terminology and findings can be compared across studies.

Strengths and Limitations

7. Limitations associated with the sample being more representative of people with parents from the Netherlands, females, seeking help from psychiatric outpatient centres, and not taking medications should be included.

Practical Implications

8. The findings are interesting and there have been some new studies coming out that have similar findings re: gatekeepers that could be referenced to complement the paper.

Reviewer #2: Summary

The manuscript provides a clear and thoughtful analysis and discussion. The authors draw a comprehensive picture of young people's access experiences, by identifying multifaceted barriers and facilitators. The diversity of the sample is a strength, and the data on participants’ educational, socio-economic, and ethnic backgrounds provide valuable context. Below are some suggestions for strengthening the manuscript, and enhancing its impact.

General Comments

Major: The abstract and introduction suggest a focus on youth who struggle to access care. However, the sample consists primarily of youth who did manage to access treatment. To set the readers’ expectations from the start, it would be helpful to emphasise that this study explored perceived facilitators and barriers, and that it did so from the perspective of youth who successfully navigated access. Given that only three out of 32 participants had never accessed mental health support, clarity and focus may be enhanced by removing these three cases from the sample.

Major: Barriers and facilitators to youth mental health treatment access are a relatively well-researched area, as demonstrated by a recently published systematic review on the topic that the authors may want to reference (Aguirre Velasco et al., 2020). The authors helpfully explain how they seek to add to existing research. It would be helpful to comment specifically on how this work adds to a study by Martínez-Hernáez et al (2014), which involved 105 in-depth interviews with depressed youth about barriers to professional help-seeking.

Major: In the discussion, the authors provide some thoughtful suggestions for how access may be facilitated, but it would be helpful to contextualise these ideas with reference to current debates and initiatives in the field, (e.g., see Aguirre Velasco et al., for a review of interventions). It appears that an implicit assumption is made that the supply of treatment resources is sufficient to meet the demand from youth who do not currently seek help. This may, however, not be the case in all contexts. Even in countries with well-developed mental health systems, waiting times are high and a major barrier to access (see Edbrooke-Childs and colleagues 2020). This could be acknowledged and discussed. Similarly, the authors could expand on what systems of care are most likely to mitigate the identified access barriers (e.g., integrated care pathways, stepped care models, needs-based care delivery).

Specific suggestions

Introduction:

*Line 52-53: (minor): “Furthermore, in this age group, depression and other mental disorders are by far the most important causes of disability (2)” – Consider providing an updated reference – e.g., the 2019 WHO fact sheet on adolescent mental health: https://www.who.int/news-room/fact-sheets/detail/adolescent-mental-health).

*Line 81-83 (minor): The authors state that quantitative research does not capture the “specific personal information, required to be able to fully understand the specific pathways of facilitating factors in help-seeking”. – Arguably, quantitative analysis can assess whether specific personal factors (e.g., demographic and clinical characteristics) predict service use. It cannot, however, provide thick narrative descriptions of how different factors or barriers influence service use. McLeod 2011 provides a thorough rationale for using qualitative research methods that may be helpful to reference.

Methods:

*(major): Line 99-100: Some additional information on the educational and mental health care institutions from which participants were recruited would be helpful. Were these high schools? Universities? Were mental health care institutions public or private? Were they outpatient or inpatient services?

*Line 122-123 (minor): “Interviews were guided by a topic list, which was developed based on both literature and expert opinion” – please elaborate what is meant by ‘expert opinion’ and what process was used to obtain this.

*Line 143 (major): “Data analysis was conducted using thematic content analysis”. Thematic analysis and content analysis may be considered to form two separate analytic approaches (see e.g. Joffe 2011). The authors reference a seminal guide to thematic analysis by Braun and Clark, and it is not clear why they call their approach thematic content analysis, rather than just thematic analysis. Clarification would be helpful. In Lines 151-156, the explanation of the coding process would benefit from review and refinement to clarify the sequence by which the authors identified overarching themes as well as more specific codes nested within each theme.

*Table 1 (minor): To protect confidentiality, the authors may want to consider grouping participants’ countries of origin. See also lines 342-346 where identification of specific countries may not be necessary (regions could be stated instead).

Results:

*Table 2 (minor): The authors state here “Not noticing problems in academic performance hindered help-seeking (b)”. The narrative discussion however suggests that some youth did not have academic performance issues, and therefore did not notice their mental health difficulties (rather than not noticing academic issues).

Discussion

*Line 408 (minor): “Individual malfunctioning – such as poor academic performance, physical symptoms, and mental distress – was often a prompt for help-seeking” – Malfunctioning may be perceived as a stigmatising term – functioning may be more neutral. Note that physical symptoms and mental distress can be distinguished conceptually from daily functioning, which refers to a young person’s ability to meet age appropriate role demands, and may be impaired by symptoms (see e.g., Rapee et al. 2012).

*Line 490 (minor): “Furthermore, a focus on adolescents and young adults aged 16-24 years is relatively uncommon” – the recent review by Aguirre Velasco, et al 2020 suggests that this broad age range has been covered by a number of studies. Consider rephrasing.

*Line 491 (minor): Typo in the reference.

*Line 492 (minor): The authors describe their qualitative sample as “extensive”. This may be misleading, as the sample is small compared with quantitative research, although sizeable for a qualitative study. Consider rephrasing.

Conclusion:

(minor) Ensure consistent wording is used to refer to the five key themes in the Abstract, Results, Discussion, and Conclusions.

References mentioned above:

Aguirre Velasco, A., Cruz, I.S.S., Billings, J. et al. What are the barriers, facilitators and interventions targeting help-seeking behaviours for common mental health problems in adolescents? A systematic review. BMC Psychiatry 20, 293 (2020). https://doi.org/10.1186/s12888-020-02659-0

Bear, H. A., Edbrooke-Childs, J., Norton, S., Krause, K. R., & Wolpert, M. (2020). Systematic Review and Meta-analysis: Outcomes of Routine Specialist Mental Health Care for Young People With Depression and/or Anxiety. Journal of the American Academy of Child & Adolescent Psychiatry, 59(7), 810–841. https://doi.org/10.1016/j.jaac.2019.12.002

Edbrooke-Childs, J., & Deighton, J. (2020). Problem severity and waiting times for young people accessing mental health services. BJPsych Open, 6(6), E118. doi:10.1192/bjo.2020.103

Joffe, H., & Yardley, L. (2011). Content and Thematic Analysis. In D. F. Marks & L. Yardley (Eds.), Research Methods for Clinical and Health Psychology. Sage Publications. https://doi.org/10.4135/9781849209793

Martínez-Hernáez, A., DiGiacomo, S.M., Carceller-Maicas, N. et al. Non-professional-help-seeking among young people with depression: a qualitative study. BMC Psychiatry 14, 124 (2014). https://doi.org/10.1186/1471-244X-14-124

McLeod, J. (2011). Qualitative Research in Counselling and Psychotherapy (2nd ed.). Sage Publications. https://doi.org/10.4135/9781849209663

Rapee, R. M., Bögels, S. M., Van Der Sluis, C. M., Craske, M. G., & Ollendick, T. (2012). Annual research review: Conceptualising functional impairment in children and adolescents. Journal of Child Psychology and Psychiatry and Allied Disciplines, 53(5), 454–468. https://doi.org/10.1111/j.1469-7610.2011.02479.x

6. PLOS authors have the option to publish the peer review history of their article (what does this mean?). If published, this will include your full peer review and any attached files.

Reviewer #1: No

Reviewer #2: No

---

## [Author Response · Author response to Decision Letter 0]

18 Dec 2020

Responses to reviewer #1:

Interview procedure

1. Were any honoraria or remuneration provided to participants?

Response: We thank the reviewer for this question. Indeed, participants received a small reward in the form of a €10 gift voucher. This has been added to the text on page 6 (under sample and recruitment): 

“Participants received a small reward in the form of a €10 gift voucher.”

Data analysis

2. How have the quotes been selected? Have quotes been select to represent a wide range of participants? The reason I ask is because the demographics table indicates a large number of participants whose parents are from the Netherlands, but many of the quotes indicate otherwise, especially in the context of the cultural influences theme. Perhaps another quote could be added in this section from someone representing the Netherlands, given the large proportion.

Response: We agree that it is important to use representative quotes. The quotes have been selected to illustrate the different themes as good as possible and to represent a wide range of participants. Especially in the context of the cultural influences, there are many quotes of participants from non-western origin. In paragraph 4.3, stigma and cultural influence (page 18), we added the following quote of a participant with Dutch parents explaining how she perceived stigma within her surroundings, to represent this part of the interviewed population:

“A participant with Dutch parents explained how she perceived stigma within her surroundings:

‘I felt ashamed about my problems and about visiting my school counsellor, especially towards my parents, but I also didn’t dare to speak about it with my peers. I think I was afraid of their judgement: ‘She can’t do it on her own, she needs help, she is weak, she is such a bad person’, you know?’ (female, age 20).“

Results

3. Qualitative is typically referred to as "findings" rather than "results"

Response: Throughout the whole manuscript we changed the word ‘results’ into ‘findings’.

4. Themes section: "personal themes" .. could you clarify what is meant by this? Perhaps this could also be clarified by the following table (2).

Response: We agree that using the term ‘personal themes’ can be confusing. We changed the text so that it becomes clear that the five main themes concern either processes within an individual (such as individual functioning, health literacy and attitudinal aspects) or externally-oriented processes (the identification of symptoms and stigma by others, and accessibility of mental health care). The text now reads as follows (under ‘themes’ page 10):

“Analysis of the interviews generated five main themes with different barriers and facilitators for help-seeking. These main themes concerned processes within individuals (individual functioning, health literacy and attitudinal aspects) and externally-oriented processes (the identification of symptoms and stigma by others, and accessibility of mental health care).” 

Additionally, we added a column in Table 2 named ‘type of process’ where we specify which themes considered ‘processes within individuals’ or considered ‘externally-oriented processes’ to give a clear overview of this division (page 10).

5. Section on physical symptoms: "a 20-year old female secondary school.." I'm wondering if this is correct given the age?

Response: This sentence is indeed confusing. This female participant appeared to be no longer in secondary school at the time of the interview, but she was talking about her experiences in the past, when she was in secondary school. We changed the text accordingly and it now reads ‘a 20-year-old female suffered from chest pains while in secondary school’ (page 12, lines 213-214). 

6. The themes could be strengthened by using (and adding to) a conceptual help-seeking framework (eg, Rickwood et al 2012) so that the terminology and findings can be compared across studies.

Response: We thank the reviewer for this valuable suggestion as a conceptual help-seeking framework indeed provides context. 

We have chosen to refer to the conceptual help-seeking framework by Rickwood & Thomas (2012) in the Methods and Discussion section, clarifying the focus of our study and thus facilitating comparison across studies. The definition of help-seeking proposed by Rickwood & Thomas (2012) was also added to our introduction: 

Introduction (page 4):

“Rickwood and Thomas (Rickwood & Thomas, 2012) proposed a general definition of help-seeking: “In the mental health context, help-seeking is an adaptive coping process that is the attempt to obtain external assistance to deal with a mental health concern”.

Methods (page 5):

“In terms of the conceptual measurement framework from Rickwood and Thomas (22), this study focused on all three Process aspects (Orientation, Intention and Behaviour), in a broad time frame (Ever), from Formal help sources (Source), in participants with a specific syndrome type, i.e. depressive symptoms (Concern). Information on all types of help (Type) was gathered, with a specific focus on information and treatment.”

Discussion (page 22):

“This qualitative study aimed to investigate facilitating factors and barriers for formal treatment-seeking in adolescents and young adults with depressive symptoms, focusing on all three Process aspects of help-seeking behaviour (Orientation, Intention and Behaviour) in a broad time frame (i.e. lifetime) (22).”

Strengths and Limitations

7. Limitations associated with the sample being more representative of people with parents from the Netherlands, females, seeking help from psychiatric outpatient centres, and not taking medications should be included.

Response: We thank the reviewer for this important remark. We used purposive sampling to include a diverse sample, resulting in a large number of non-Western participants from various backgrounds (37.5%) and a proportion of males (34.4%) representative of the population (adolescents with depressive symptoms). Participants from outpatient centres and to a lesser extent drug-naïve participants are relatively overrepresented.

We have altered the text in the Strengths and Limitations section: on page 26 accordingly:

“A limitation of this study is the relatively small number of low-educated adolescents compared to high-educated participants. In addition, the number of untreated participants was relatively small, despite efforts to include more participants without a treatment history in the study. This could have resulted in an underrepresentation of barriers and facilitators that are specific to these subgroups. The findings of this study are most representative for females, adolescents from Dutch descent, drug-naïve adolescents and adolescents (eventually) seeking help from psychiatric outpatient centres. Future research could specifically focus on adolescents that did not receive professional help and low-educated adolescents to explore whether other processes play a role within the help-seeking process of these specific subgroups.” 

Practical Implications

8. The findings are interesting and there have been some new studies coming out that have similar findings re: gatekeepers that could be referenced to complement the paper.

Response: In accordance to your suggestions we have added recent literature to both the introduction and the discussion. To the introduction we added 7 recent studies in the part where we discuss previous research on barriers and facilitators in help-seeking behavior for mental health problem (page 4). Most importantly, we added the recent systematic review on facilitators and barriers in help-seeking behaviour of adolescents with mental health problems and suggested interventions, as suggested by reviewer 2 (Aguirre Velasco et al., 2020) (page 4).The text now reads as follows:

“A recent systematic review including 54 studies on barriers and facilitators in help seeking behaviours for common mental health problems in adolescents found that the two most cited barriers were stigma and negative beliefs towards mental health services and professionals (23). Research also shows that the desire to handle problems on one’s own (24,25), low perceived need for help (20,25,26),difficulty in identifying symptoms of mental illness (26,27), perceived fear of psychotherapy, the belief that a psychotherapist would not be able to be found and financial concerns were also barriers to seeking help in adolescents (28). While research into barriers to treatment for adolescents is readily available, few studies have addressed the role of facilitating factors. The little research that has been done shows that mental health literacy (29,30), positive past experiences with help-seeking (31–35), social support or encouragement from others (36,37) and confidentiality and trust in the provider (25,38) might facilitate help-seeking for adolescents with mental health issues. However, research on facilitating factors is scarce (23).”

Throughout the discussion, we included 2 studies (Aguirre Velasco et al., 2020; Martínez-Hernáez et al., 2014) in the sections on comparison with the literature (page 23, lines 456;464;469).

We highlighted newly added references in the revised manuscript. 

Responses to reviewer #2:

9. Major: The abstract and introduction suggest a focus on youth who struggle to access care. However, the sample consists primarily of youth who did manage to access treatment. To set the readers’ expectations from the start, it would be helpful to emphasise that this study explored perceived facilitators and barriers, and that it did so from the perspective of youth who successfully navigated access. Given that only three out of 32 participants had never accessed mental health support, clarity and focus may be enhanced by removing these three cases from the sample.

Response: The reviewer addresses an important point, and we acknowledge that our sample primarily consists of participants who (eventually) accessed formal care. To set the readers’ expectations from the beginning we added a description of the sample to the abstract and we rephrased a sentence in the introduction to clarify that we focused on the search for professional help. The text now reads as follows:

Abstract: 

“Our sample consisted mainly of adolescents who eventually found their way to professional help.” (page 2)

Introduction: 

“The aim of the current qualitative study is to investigate which facilitating factors and barriers play a role in the search for professional help of depressed adolescents and young adults aged 16 to 24 years, using in-depth semi-structured individual interviews.” (page 5)

In the Methods section we now refer to the help-seeking framework of Rickwood and Thomas (2012) which clarifies the focus of our study (see Reviewer 1, point 6). The underrepresentation of adolescents who did not receive help is a limitation in our study and we now elaborate on this in greater detail in the discussion (see Reviewer 1, point 7). Although we do understand the reviewer’s suggestion, after discussion in our research group we decided not to remove the 3 participants who did not receive help from our sample, because a) we think they are a valuable addition to our findings because they bring a different, valuable, perspective, b) the inclusion criteria for our study and research question was focused on adolescents with all degrees of depressive symptomatology, who received help or who did not (yet) receive help. Unfortunately our sample was not as diverse on this point as we would have hoped, on which we reflect in the discussion. Further research on this specific subgroup is important, and we added a recommendation on this subject to the discussion (see Reviewer 1, point 7). 

10. Major: Barriers and facilitators to youth mental health treatment access are a relatively well-researched area, as demonstrated by a recently published systematic review on the topic that the authors may want to reference (Aguirre Velasco et al., 2020). The authors helpfully explain how they seek to add to existing research. It would be helpful to comment specifically on how this work adds to a study by Martínez-Hernáez et al (2014), which involved 105 in-depth interviews with depressed youth about barriers to professional help-seeking.

Response: We would like to thank the reviewer for these valuable suggestions. We added the recent systematic review of Aguirre Velasco et al. (2020) to the literature in our introduction. We also added the article of Martínez-Hernáez et al. (2014) to the introduction to complete the overview of previous research on this topic. For specifications on how the text has been changed, see point 8 of reviewer #1.

Our study adds to the article of Martínez-Hernáez et al. by not only focusing on barriers but also (specifically) on facilitators of the help-seeking process. Insight into facilitators provides additional information on what can be improved in adolescents’ and young adults’ access to mental health care. As suggested, we added a comparison to the article of Martínez-Hernáez et al. (2014) in our Discussion:

Discussion (page 22):

“This study adds to the literature because contrary to previous studies focusing primarily on barriers (24–26,42,43) there was specific attention for facilitating factors, yielding valuable additional information on adolescents’ and young adults’ pathways to professional mental health care.”

Strengths & Limitations (page 25):

“Furthermore, in addition to previous qualitative studies focussing on barriers (24–26,42,43), our study paid specific attention to facilitators of the help-seeking process as well.”

11. Major: In the discussion, the authors provide some thoughtful suggestions for how access may be facilitated, but it would be helpful to contextualise these ideas with reference to current debates and initiatives in the field, (e.g., see Aguirre Velasco et al., for a review of interventions). It appears that an implicit assumption is made that the supply of treatment resources is sufficient to meet the demand from youth who do not currently seek help. This may, however, not be the case in all contexts. Even in countries with well-developed mental health systems, waiting times are high and a major barrier to access (see Edbrooke-Childs and colleagues 2020). This could be acknowledged and discussed. Similarly, the authors could expand on what systems of care are most likely to mitigate the identified access barriers (e.g., integrated care pathways, stepped care models, needs-based care delivery).

Response: We appreciate this recommendation. Part of the section on ‘Practical implications’ has been rewritten according to the reviewers’ suggestions: (page 26-27): 

“Previous research has found promising effects from education and awareness programmes on mental health literacy, help-seeking behaviour or intention to seek help and stigma reduction (59–63), using different methods including psycho-education and (video) presentations with former patients.” (page 26)

“Peer leading interventions have been developed previously and shown a positive effect (61) on referring suicidal peers for adult support.” (page 27)

“Furthermore, in our sample waiting time appeared to be a problem. This is in accordance with previous research (62,63) and indicates that the number of young adults in need of treatment exceeds the available treatment resources. This problem needs particular attention from policy makers and governments, and suggests that rearrangement of treatment resources for youngsters might be necessary. One of the most promising approaches to facilitate access to care seems to be the provision of basic mental health care in schools (58,64). This could be considered a form of stepped care, where school counselors and psychologists function as primary support for distressed adolescents.” (page 27)

Specific suggestions

Introduction:

12. *Line 52-53: (minor): “Furthermore, in this age group, depression and other mental disorders are by far the most important causes of disability (2)” – Consider providing an updated reference – e.g., the 2019 WHO fact sheet on adolescent mental health: https://www.who.int/news-room/fact-sheets/detail/adolescent-mental-health).

Response: We thank the reviewer for this suggestion. We updated this reference in our manuscript (page 3, line 55).

13. *Line 81-83 (minor): The authors state that quantitative research does not capture the “specific personal information, required to be able to fully understand the specific pathways of facilitating factors in help-seeking”. – Arguably, quantitative analysis can assess whether specific personal factors (e.g., demographic and clinical characteristics) predict service use. It cannot, however, provide thick narrative descriptions of how different factors or barriers influence service use. McLeod 2011 provides a thorough rationale for using qualitative research methods that may be helpful to reference.

Response: We agree and have rephrased this sentence. We also strengthened our rationale for qualitative research with the reference of McLeod, 2011.The new text is as follows (page 4):

“A downside to these research designs might be that narrative descriptions, required to be able to fully understand the specific pathways of facilitating factors in help-seeking, might be missed. Qualitative research is specifically intended to promote the growth of understanding, rather than to collect factual knowledge and causal explanations (McLeod, 2011). No previous research uses in-depth individual interviewing to study barriers and specifically facilitators within the help-seeking process of adolescents with depressive symptomatology.”

Methods:

14. *(major): Line 99-100: Some additional information on the educational and mental health care institutions from which participants were recruited would be helpful. Were these high schools? Universities? Were mental health care institutions public or private? Were they outpatient or inpatient services?

Response: We agree with the reviewer and understand that the Dutch school system may be confusing for readers from other countries, and hence, that a clear description of educational attainment is required. However, direct comparison of the Dutch educational system to other (e.g. British or American) systems is challenging. In our sample, multiple MBO’s were approached for inclusion, which is vocational education at various levels (varying from 1 to 4 years duration). Higher education in The Netherlands can be categorized into two types of institutions, so-called ‘hogescholen’ and universities. The former include education in a particular field with a practical focus, whereas universities are academic institutions. More information can also be found on https://en.wikipedia.org/wiki/Education_in_the_Netherlands. 

The affiliated mental health care institutions were public, and patients were recruited from outpatient settings only. 

More detailed information on both the educational and mental health care institutions from which participants were recruited is added to the Methods section in order to provide the necessary clarification, on page 5 :

“Educational institutions included secondary vocational schools and three institutions for higher education of which one university. Participating mental health care institutions were public, outpatient mental health care centres.”

15. *Line 122-123 (minor): “Interviews were guided by a topic list, which was developed based on both literature and expert opinion” – please elaborate what is meant by ‘expert opinion’ and what process was used to obtain this.

Response: We appreciate this remark from the reviewer and would like to clarify the process. After a draft version of the topic list was created using relevant literature, this draft version was discussed with experts from the field (including mental health care professionals and an educational counsellor, the research group and a client panel of two patients) for further fine-tuning and was adjusted in accordance with their remarks. The topic list was further updated during the process of data collection. 

An clarification of the process has been added to the Methods section, on page 6:

“Interviews were guided by a topic list, which was developed based on both literature and expert opinion. After a draft version of the topic list was created using relevant literature, it was discussed with experts from the field (including mental health care professionals, an educational counsellor, the research group and a client panel of two patients) for further fine-tuning and was adjusted accordingly.”

16. *Line 143 (major): “Data analysis was conducted using thematic content analysis”. Thematic analysis and content analysis may be considered to form two separate analytic approaches (see e.g. Joffe 2011). The authors reference a seminal guide to thematic analysis by Braun and Clark, and it is not clear why they call their approach thematic content analysis, rather than just thematic analysis. Clarification would be helpful. In Lines 151-156, the explanation of the coding process would benefit from review and refinement to clarify the sequence by which the authors identified overarching themes as well as more specific codes nested within each theme.

Response: We understand the confusion of the reviewer regarding this point, and realize the text is not clear. The reviewer is correct that we reference Braun and Clarke (2006) who use the term thematic analysis, and this has been altered in the text on page (7). Different authors use different terminology (e.g. ’thematic content analysis’ is used by Green & Thorogood) which might have caused confusion. 

Furthermore, more detailed information on the coding process has been added to the Methods section. The first interviews were carefully read and then manually coded by two independent interviewers. Differences were discussed until consensus was reached. Then, a preliminary thematic map was developed based on these first interviews and discussed in a small research team (RW, MvM, AM). The thematic map was further updated and adapted after every two or three new interviews in an iterative process by four of the researchers. The main themes from the initial thematic map were then discussed amongst the coders and the research team and further reviewed and adjusted in subsequent meetings resulting in a final thematic map with the main themes. We added the following text to the Methods section on page 8: 

“A preliminary thematic map was developed by two interviewers (RW, MvM) based on the first independently coded interviews and discussed in a small research team (RW, MvM, AM). The thematic map was further updated and adapted after every two or three new interviews in an iterative process by four of the interviewers (RW, EE, MvM, AIM). The main themes from the initial thematic map, differences and similarities between cases and possible explanations were then discussed amongst the coders and the research team (consisting of two psychiatrists (AvB, NB), one qualitative researcher (MW), one psychologist (AM) plus the aforementioned interviewers), and further reviewed and adjusted in subsequent meetings resulting in a final thematic map including the main themes..”

17. *Table 1 (minor): To protect confidentiality, the authors may want to consider grouping participants’ countries of origin. See also lines 342-346 where identification of specific countries may not be necessary (regions could be stated instead).

Response: We thank the reviewer for this thoughtful suggestion.To protect confidentiality we regrouped participants by geographical subregion (in line with the United Nations geoscheme). In Table 1 and in paragraph 4.3 (Stigma and cultural influences, page 18) all countries have been removed and replaced by geographical subregion.

Results:

18. *Table 2 (minor): The authors state here “Not noticing problems in academic performance hindered help-seeking (b)”. The narrative discussion however suggests that some youth did not have academic performance issues, and therefore did not notice their mental health difficulties (rather than not noticing academic issues).

Response: We thank the reviewer for noticing this and we now see this is not clear from our text. To be complete we changed the text in Table 2 (page 10) to “Not noticing or not experiencing problems in academic performance hindered help-seeking (b)”.

Discussion

19. *Line 408 (minor): “Individual malfunctioning – such as poor academic performance, physical symptoms, and mental distress – was often a prompt for help-seeking” – Malfunctioning may be perceived as a stigmatising term – functioning may be more neutral. Note that physical symptoms and mental distress can be distinguished conceptually from daily functioning, which refers to a young person’s ability to meet age appropriate role demands, and may be impaired by symptoms (see e.g., Rapee et al. 2012).

Response: We agree with the reviewer and changed ‘individual malfunctioning’ in ‘individual functioning’. Furthermore, we replaced the name of the main theme ‘individual functioning’ by ‘individual functioning and well-being’ throughout the manuscript to more adequately reflect the content of this theme. 

20. *Line 490 (minor): “Furthermore, a focus on adolescents and young adults aged 16-24 years is relatively uncommon” – the recent review by Aguirre Velasco, et al 2020 suggests that this broad age range has been covered by a number of studies. Consider rephrasing.

Response: Taking into account the reviewer’s remark and the recent review by Aguirre Velasco et al., we removed this sentence from the ‘Strengths & Limitations’ section (page 25).

21. *Line 491 (minor): Typo in the reference.

Response: Since this sentence was removed from the text (see point 20), the typo was removed as well. 

22. *Line 492 (minor): The authors describe their qualitative sample as “extensive”. This may be misleading, as the sample is small compared with quantitative research, although sizeable for a qualitative study. Consider rephrasing.

Response: We agree with the reviewer and rephrased this sentence (page 25): 

“Another strength of the study is the, for qualitative research standards, sizeable sample consisting of 32 participants who were recruited in various contexts” 

Conclusion:

23. (minor) Ensure consistent wording is used to refer to the five key themes in the Abstract, Results, Discussion, and Conclusions.

Response: We made sure that the words we used for the main themes were consistent throughout the manuscript. This resulted in consistently referring to the five key themes in the same way and order as in this part of the abstract (page 2): 

“Five main themes in help-seeking by adolescents and young adults were identified: (I) Individual functioning and well-being, (II) Health literacy, (III) Attitudinal aspects, (IV) Surroundings, and (V) Accessibility.” 

References

Aguirre Velasco, A., Cruz, I. S. S., Billings, J., Jimenez, M., & Rowe, S. (2020). What are the barriers, facilitators and interventions targeting help-seeking behaviours for common mental health problems in adolescents? A systematic review. BMC Psychiatry, 20(1), 1–22. https://doi.org/10.1186/s12888-020-02659-0

Braun, V., & Clarke, V. (2006). Using thematic analysis in psychology. Qualitative Research in Psychology, 3(2), 77–101. https://doi.org/10.1191/1478088706qp063oa

King K, Strunk C, S. M. (2011). Preliminary effectiveness of surviving the teens® suicide prevention and depression awareness program on adolescents’ Suicidality and self-efficacy in performing help-seeking behaviors. J School Health Am School Health Assoc., 81, 581–590.

Martínez-Hernáez, A., DiGiacomo, S. M., Carceller-Maicas, N., Correa-Urquiza, M., & Martorell-Poveda, M. A. (2014). Non-professional-help-seeking among young people with depression: A qualitative study. BMC Psychiatry, 14, 1–11. https://doi.org/10.1186/1471-244X-14-124

McLeod, J. (2011). Qualitative research in counselling and psychotherapy. SAGE Publications Ltd. https://doi.org/10.4135/9781849209663

Rickwood D, Cavanagh S, Curtis L, S. R. (2004). Educating young people about mental health and mental illness: evaluating a school-based programme. Int J Ment Health Promot, 6(4), 23–32.

Rickwood, D., & Thomas, K. (2012). Conceptual measurement framework for help-seeking for mental health problems. Psychology Research and Behavior Management, 5, 173–183. https://doi.org/10.2147/PRBM.S38707

Robinson J, Gook S, Yuen HP, Hughes A, Dodd S, Bapat S, Y. A. (2010). Depression education and identification in schools: an Australian-based study. Sch Ment Heal, 2, 13–22.

Saporito, J. M. (2009). Reducing stigma toward seeking mental health treatment. Dissertation Abstracts International: Section B: The Sciences and Engineering, 70(6-B), 3794.

Strunk CM, Sorter MT, Ossege J, K. K. (2014). Emotionally troubled teens’ help- seeking behaviors: an evaluation of surviving the teens (R) suicide prevention and depression awareness program. J Sch Nurs, 30(5), 366–375. https://doi.org/https://doi.org/10.1177/1059840513511494

Wyman P, Hendricks Brown C, LoMurray M, Schmeelk-Cone K, Petrova M, Yu Q, Walsh E, Tu X, W. W. (2010). An outcome evaluation of the sources of strength suicide prevention program delivered by adolescent peer leaders in high schools. American Journal of Public Health, 100(9), 1653–1661.

---

## [Decision Letter · Decision Letter 1]

14 Jan 2021

PONE-D-20-29462R1

Facilitating factors and barriers in help-seeking behaviour in adolescents and young adults with depressive symptoms: a qualitative study

PLOS ONE

Dear Dr. Eigenhuis,

Thank you for submitting your manuscript to PLOS ONE. After careful consideration, we feel that it has merit but does not fully meet PLOS ONE’s publication criteria as it currently stands. Therefore, we invite you to submit a revised version of the manuscript that addresses the points raised during the review process. One of the reviewers still has some outstanding questions that need to be answered.

We look forward to receiving your revised manuscript.

Kind regards,

Therese van Amelsvoort

Academic Editor

PLOS ONE

Reviewers' comments:

Reviewer's Responses to Questions

**Comments to the Author**

1. If the authors have adequately addressed your comments raised in a previous round of review and you feel that this manuscript is now acceptable for publication, you may indicate that here to bypass the “Comments to the Author” section, enter your conflict of interest statement in the “Confidential to Editor” section, and submit your "Accept" recommendation.

Reviewer #2: (No Response)

2. Is the manuscript technically sound, and do the data support the conclusions?

Reviewer #2: Partly

3. Has the statistical analysis been performed appropriately and rigorously? 

Reviewer #2: N/A

4. Have the authors made all data underlying the findings in their manuscript fully available?

Reviewer #2: Yes

5. Is the manuscript presented in an intelligible fashion and written in standard English?

Reviewer #2: Yes

6. Review Comments to the Author

Reviewer #2: I thank the authors for their thorough and helpful response letter. The discussion of additional research in the introduction is helpful, and the additions to the limitations and implications sections strengthen the paper. However, I have some concerns about the conceptual integrity and clarity of the paper as currently presented, and believe further clarifications are needed.

Reviewer #1 suggested that “The [five help-seeking] themes could be strengthened by using (and adding to) a conceptual help-seeking framework (eg, Rickwood et al 2012) so that the terminology and findings can be compared across studies.”

In response to this suggestion, the authors now state that the study was informed by the help-seeking framework by Rickwood and Thomas (2012):

“In terms of the conceptual measurement framework from Rickwood and Thomas (22), this study focused on all three Process aspects (Orientation, Intention and Behaviour), in a broad time frame (Ever), from Formal help sources (Source), in participants with a specific syndrome type, i.e. depressive symptoms (Concern). Information on all types of help (Type) was gathered, with a specific focus on information and treatment.”

However, it remains unclear how exactly the framework did inform the paper. The authors state that the study focused “on all three Process aspects of help-seeking behaviour (Orientation, Intention and Behaviour)” (p. 22, line 442), but there is no explicit discussion of these three process aspects either in the findings or discussion section, and they also do not appear in the coding frame (Table 2). More generally, it does not seem that the authors used the framework to structure the discussion of findings, or their contextualisation within existing literature, as suggested by reviewer #1. As such, the Rickwood framework appears to be an afterthought, rather than a conceptual backbone. I might suggest that the authors engage with the framework more thoroughly and meaningfully, in line with the suggestions made by Reviewer #1. Alternatively, the authors could consider removing reference to the framework altogether. I might avoid a middle ground that may appear tokenistic and could undermine the paper’s credibility. If the authors decide to maintain reference to the conceptual framework, it would be helpful to (a) introduce its key constructs and dimensions in the introduction, (b) clarify that it did not inform the study design (e.g., design of the topic guide; design of the coding frame), (c) state more explicitly how it informed the paper.

In the Methods, the authors have helpfully expanded on the coding process. In addition, it would be helpful to state explicitly whether the coding process and creation of the thematic map was inductive, or informed by an existing coding frame (such as the framework by Rickwood and Thomas for example).

At the start of the findings section, it might be helpful to introduce the five key help-seeking themes identified by the authors, rather than focusing just on the distinction between processes centred within the individual and externally focused processes. It is not entirely clear what value this additional taxonomic layer adds, in addition to the five help-seeking themes, and how this layer relates to the process aspects described by the Rickwood and Thomas framework (i.e., Orientation, Intention and Behaviour). Clarification would be helpful.

Table 3 contains reference to Moroccan parents. The authors may want to adjust to “North Africa” for consistency with the remainder of the paper.

In several parts of the discussion (twice on page 23), the authors now state that “individual functioning and wellbeing – such as poor academic performance, physical symptoms, and mental distress – was often a prompt for help-seeking". The authors may want to clarify that it was impairment or a deterioration in functioning and wellbeing that prompted help-seeking.

7. PLOS authors have the option to publish the peer review history of their article (what does this mean?). If published, this will include your full peer review and any attached files.

Reviewer #2: No

---

## [Author Response · Author response to Decision Letter 1]

5 Feb 2021

Prof. dr. T. van Amelsvoort

Academic Editor

PLOS ONE

February 5, 2021

Subject: Revision “Facilitating factors and barriers in help-seeking behaviour in adolescents and young adults with depressive symptoms: a qualitative study” (PONE-D-20-29462R1).

Dear prof. dr. van Amelsvoort,

Thank you for reviewing our manuscript and for the opportunity to submit a revision. We appreciate the feedback from Reviewer #2 and have taken it into consideration. 

Although we intend to comply with the requests of Reviewer #1 and Reviewer #2, we feel that the remarks about the integration of the conceptual framework are difficult to address in such a way that does justice to the suggestions and concerns of both reviewers. 

After deliberation in our research group, we have chosen to integrate the framework as described in the Response to reviewers below. Hopefully, you will appreciate our proposed revision. 

We believe that the manuscript has improved by incorporating the reviewers’ suggestions and hope you will consider it for publication in PLOS ONE.

Please feel free to contact us in case further clarifications are needed.

On behalf of all co-authors,

Yours sincerely,

Eline Eigenhuis and Ruth C. Waumans 

Amsterdam UMC, location VUmc, Department of Psychiatry

GGZ inGeest 

Oldenaller 1, 1081 HJ Amsterdam, The Netherlands. 

E-mail: e.eigenhuis@ggzingeest.nl

 

Response to Reviewer #2:

Reviewer #2: I thank the authors for their thorough and helpful response letter. The discussion of additional research in the introduction is helpful, and the additions to the limitations and implications sections strengthen the paper. However, I have some concerns about the conceptual integrity and clarity of the paper as currently presented, and believe further clarifications are needed.

Reviewer #1 suggested that “The [five help-seeking] themes could be strengthened by using (and adding to) a conceptual help-seeking framework (eg, Rickwood et al 2012) so that the terminology and findings can be compared across studies.”

In response to this suggestion, the authors now state that the study was informed by the help-seeking framework by Rickwood and Thomas (2012):

“In terms of the conceptual measurement framework from Rickwood and Thomas (22), this study focused on all three Process aspects (Orientation, Intention and Behaviour), in a broad time frame (Ever), from Formal help sources (Source), in participants with a specific syndrome type, i.e. depressive symptoms (Concern). Information on all types of help (Type) was gathered, with a specific focus on information and treatment.”

However, it remains unclear how exactly the framework did inform the paper. The authors state that the study focused “on all three Process aspects of help-seeking behaviour (Orientation, Intention and Behaviour)” (p. 22, line 442), but there is no explicit discussion of these three process aspects either in the findings or discussion section, and they also do not appear in the coding frame (Table 2). More generally, it does not seem that the authors used the framework to structure the discussion of findings, or their contextualisation within existing literature, as suggested by reviewer #1. As such, the Rickwood framework appears to be an afterthought, rather than a conceptual backbone. I might suggest that the authors engage with the framework more thoroughly and meaningfully, in line with the suggestions made by Reviewer #1. Alternatively, the authors could consider removing reference to the framework altogether. I might avoid a middle ground that may appear tokenistic and could undermine the paper’s credibility. If the authors decide to maintain reference to the conceptual framework, it would be helpful to (a) introduce its key constructs and dimensions in the introduction, (b) clarify that it did not inform the study design (e.g., design of the topic guide; design of the coding frame), (c) state more explicitly how it informed the paper.

Response: We thank the reviewer for this important feedback. We endorse the remark from the reviewer that the framework has not been used to structure our findings or to form the coding tree, and as such was not part of the design of our study. We do however value the suggestion of Reviewer #1 to use a conceptual framework to structure findings in studies on help-seeking. After thorough discussion within our research group on how to follow up on the suggestions of both reviewers, we came to the conclusion that integrating the framework in this phase of the study does not match with the inductive design of our study. Therefore, we decided to remove the text about the framework from the Methods section. 

However, we acknowledge the importance of conceptual consistency in order to enable comparison of findings across studies using a conceptual framework such as proposed by Rickwood and Thomas. The following text has therefore been added to the Discussion section (page 26): 

“Previous research has shown that the literature on help-seeking behaviour comprises a variety of studies with different methodology, focus and outcomes, hampering direct comparison (22). In order to improve conceptual consistency, Rickwood and Thomas (22) presented a framework consisting of five aspects of help-seeking behaviour: Process (comprising the aspects of Orientation, Intention, Behaviour), Timeframe (i.e. past/next 4 weeks, past/next 12 months, Ever), Source (Formal help, Semi-formal help, Informal help, Self-Help), Type of help (Instrumental, Information, Affiliative, Emotional, Treatment) and Concern (General distress, Specific symptom types). Although we did not include this framework in the design of our study, our study can be categorised as focusing on all three Process aspects (Orientation, Intention and Behaviour) of help-seeking, in a broad time frame (Ever), from Formal help sources (Source), in participants with a specific syndrome type, i.e. depressive symptoms (Concern). Information on all types of help (Type) was gathered, with a specific focus on information and treatment. Future studies on help-seeking may use a conceptual framework such as described by Rickwood and Thomas (22) to increase homogeneity in study designs and comparability of the results.”

In the Methods, the authors have helpfully expanded on the coding process. In addition, it would be helpful to state explicitly whether the coding process and creation of the thematic map was inductive, or informed by an existing coding frame (such as the framework by Rickwood and Thomas for example).

Response: In order to clarify the process of data analysis, the text in the Methods section has been altered (page 7):

“Data analysis was conducted in an inductive manner using thematic analysis (41), focusing on participants’ perceived barriers and facilitators in treatment-seeking.”

At the start of the findings section, it might be helpful to introduce the five key help-seeking themes identified by the authors, rather than focusing just on the distinction between processes centred within the individual and externally focused processes. It is not entirely clear what value this additional taxonomic layer adds, in addition to the five help-seeking themes, and how this layer relates to the process aspects described by the Rickwood and Thomas framework (i.e., Orientation, Intention and Behaviour). Clarification would be helpful.

Response: We agree with the reviewer that the focus of the start of the results section should be on the five main themes and their introduction. The distinction between internal and external processes was therefore removed from our text and from Table 2. The text now reads (page 10):

“Analysis of the interviews generated five main themes with different barriers and facilitators for help-seeking. These main themes were (I) Individual functioning and well-being, (II) Health literacy, (III) Attitudinal aspects, (IV) Surroundings, and (V) Accessibility. A complete overview of the findings can be found in Table 2. The most important themes are explained below.” 

Table 3 contains reference to Moroccan parents. The authors may want to adjust to “North Africa” for consistency with the remainder of the paper.

Response: We thank the reviewer for noticing this inconsistency. We changed the text in Table 3 to ‘parents originating from Northern Africa’. 

In several parts of the discussion (twice on page 23), the authors now state that “individual functioning and wellbeing – such as poor academic performance, physical symptoms, and mental distress – was often a prompt for help-seeking". The authors may want to clarify that it was impairment or a deterioration in functioning and wellbeing that prompted help-seeking.

Response: We agree that the text could be clarified at this point. We added ‘impairment or a deterioration’ in two sentences of the discussion. The text now reads as follows (pages 22 & 23):

“Impairment or deterioration in individual functioning and well-being– such as poor academic performance, physical symptoms, and mental distress – was often a prompt for help-seeking, whereas good academic performance despite depressive symptoms could be a hindrance.” 

And:

“Our finding that impairment or deterioration in individual functioning and well-being prompt treatment-seeking is in accordance with previous research in adolescents (24,32,44)”

---

## [Decision Letter · Decision Letter 2]

9 Feb 2021

Facilitating factors and barriers in help-seeking behaviour in adolescents and young adults with depressive symptoms: a qualitative study

PONE-D-20-29462R2

Dear Dr. Eigenhuis,

We’re pleased to inform you that your manuscript has been judged scientifically suitable for publication and will be formally accepted for publication once it meets all outstanding technical requirements.

Kind regards,

Therese van Amelsvoort

Academic Editor

PLOS ONE

Additional Editor Comments (optional):

Reviewers' comments:

Reviewer's Responses to Questions

**Comments to the Author**

1. If the authors have adequately addressed your comments raised in a previous round of review and you feel that this manuscript is now acceptable for publication, you may indicate that here to bypass the “Comments to the Author” section, enter your conflict of interest statement in the “Confidential to Editor” section, and submit your "Accept" recommendation.

Reviewer #2: All comments have been addressed

2. Is the manuscript technically sound, and do the data support the conclusions?

Reviewer #2: Yes

3. Has the statistical analysis been performed appropriately and rigorously? 

Reviewer #2: N/A

4. Have the authors made all data underlying the findings in their manuscript fully available?

Reviewer #2: Yes

5. Is the manuscript presented in an intelligible fashion and written in standard English?

Reviewer #2: Yes

6. Review Comments to the Author

Reviewer #2: Thank you for the chance to review this manuscript one more time. The authors' decision to discuss the Rickwood and Thomas framework in the discussion section works well in my opinion, and strengthens the paper. I have no further comments and wish the authors well for disseminating their work.

7. PLOS authors have the option to publish the peer review history of their article (what does this mean?). If published, this will include your full peer review and any attached files.

Reviewer #2: No

---

## [Editor Report · Acceptance letter]

12 Feb 2021

PONE-D-20-29462R2 

Facilitating factors and barriers in help-seeking behaviour in adolescents and young adults with depressive symptoms: a qualitative study 

Dear Dr. Eigenhuis:

I'm pleased to inform you that your manuscript has been deemed suitable for publication in PLOS ONE. Congratulations! Your manuscript is now with our production department. 

Kind regards, 

on behalf of

Prof. Therese van Amelsvoort 

Academic Editor

PLOS ONE